# Emerging Effects of IL-33 on COVID-19

**DOI:** 10.3390/ijms232113656

**Published:** 2022-11-07

**Authors:** Yuan Gao, Luwei Cai, Lili Li, Yidan Zhang, Jing Li, Chengliang Luo, Ying Wang, Luyang Tao

**Affiliations:** Department of Forensic Science, Medical School of Soochow University, Add 178 Ganjiang East Road, Suzhou 215123, China

**Keywords:** COVID-19, IL-33, neutrophils, innate lymphocytes, dendritic cells, macrophages, CD4^+^ T cells, Th17/Treg cells, CD8^+^ T cells

## Abstract

Since the start of COVID-19 pandemic caused by severe acute respiratory syndrome coronavirus 2 (SARS-CoV-2), more than 6 million people have lost their lives worldwide directly or indirectly. Despite intensified efforts to clarify the immunopathology of COVID-19, the key factors and processes that trigger an inflammatory storm and lead to severe clinical outcomes in patients remain unclear. As an inflammatory storm factor, IL-33 is an alarmin cytokine, which plays an important role in cell damage or infection. Recent studies have shown that serum IL-33 is upregulated in COVID-19 patients and is strongly associated with poor outcomes. Increased IL-33 levels in severe infections may result from an inflammatory storm caused by strong interactions between activated immune cells. However, the effects of IL-33 in COVID-19 and the underlying mechanisms remain to be fully elucidated. In this review, we systematically discuss the biological properties of IL-33 under pathophysiological conditions and its regulation of immune cells, including neutrophils, innate lymphocytes (ILCs), dendritic cells, macrophages, CD4^+^ T cells, Th17/Treg cells, and CD8^+^ T cells, in COVID-19 phagocytosis. The aim of this review is to explore the potential value of the IL-33/immune cell pathway as a new target for early diagnosis, monitoring of severe cases, and clinical treatment of COVID-19.

## 1. Introduction

COVID-19, caused by severe acute respiratory syndrome coronavirus 2 (SARS-CoV-2), has a long incubation period and is highly contagious [1,2]. Since the start of this international public health emergency, there have been more than 400 million confirmed cases, with more than 6 million deaths worldwide. Patients with COVID-19 can be clinically classified into four categories: mild, common, severe, and critical. The most common clinical manifestations are fever, cough, shortness of breath, muscle aches, confusion, headache, sore throat, runny nose, chest pain, diarrhea, nausea, and vomiting [3]. The pathological process of COVID-19 can be divided into three stages: (1) SARS-CoV-2 invasion and replication; (2) immune response and cytokine storm; (3) acute respiratory distress syndrome and multiple organ dysfunction syndromes [4]. The exact pathogenesis of novel coronavirus pneumonia is not fully understood, although the secondary systemic immune cell inflammatory response and cytokine storm caused by SAR-Cov-2 have been recognized as an important cause of its high lethality [5]. Recent studies have shown that SAR-CoV-2 viral RNA is sensed by the intrinsic immune system, which generates a rapid antiviral cascade that causes uncontrolled and massive release of cytokines (e.g., IL-1β, IL-6, TNF-α, and IL-33) from infected epithelial cells, endothelial cells, and immune cells. These events initiate an uncontrolled systemic inflammatory response, ultimately leading to multi-organ failure (Table 1) [6]. Despite intensified efforts to unravel the immunopathology of COVID-19, there is still a lack of effective measures for its prevention and treatment due to insufficient understanding of the pathological mechanism.

Vascular endothelial cell damage, excessive immunity, and cross-talk are important factors in the pathogenesis and exacerbation of COVID-19 [5]. Based on the importance of the inflammatory response in the development of COVID-19, targeting and controlling pathological hyperinflammation may be an effective therapeutic measure against neocrown pneumonia. IL-33, a crucial inflammatory cytokine, can not only enhance the accumulation of neutrophil extracellular traps, promote the transformation of macrophages to the M2 type, inhibit the function of plasmacytoid dendritic cells to produce type I interferon, and drive type 2 innate lymphocytes (ILC2) differentiation, but it can also promote the proliferation of Th17 cells and limit the function of CD4^+^/CD8^+^ cells during the pathogenesis and development of COVID-19. Clinical trials have shown significantly elevated levels of serum cytokines, including IL-33 and ST2, in 100 COVID-19 subjects, and elevated levels of IL-33 and its specific receptor ST2 were identified as an independent predictor of poor prognosis for SARS-CoV-2 patients aged < 70 years, indicating that IL-33 is very likely to be involved in the pathological process of COVID-19. However, the specific effects of IL-33 in COVID-19 and the underlying mechanisms remain elusive. Here, we provide a comprehensive overview of the biological characteristics of IL-33 and its roles in the physiology and pathology of COVID-19, as well as the potential mechanisms, with the aim of exploring the value of IL-33 as a new target for the prevention and treatment of COVID-19.

## 2. Molecular Biology of IL-33

### 2.1. Molecular Structure and Function of IL-33

Interleukin 33 (IL-33) protein was first discovered in 2003 by Baekkevold et al. [7] as a nuclear protein expressed in high endothelial venule-specialized blood vessels, where it mediated the trafficking of lymphocytes to lymph nodes and secondary lymphoid tissues and was named as nuclear factor derived from high endothelial venules (NF-HEV) accordingly [8]. Schmitz et al. later showed that the IL-33 gene sequence has a high degree of homology with the gene sequences of IL-1 family members IL-1β and IL-18 [9]. The human and mouse IL-33 genes are located on chromosome 9 (9p24.1) and chromosome 19 (19C1), respectively [10,11]. In both cases, the full-length IL-33 gene includes eight coding exons, which are transcribed and translated to produce proteins containing 270 and 266 amino acids (AAs) (Figure 1A). These two proteins have 55% homology and molecular weights of approximately 30 and 29.9 kDa, respectively [12]. IL-33 is composed of an N-terminal nuclear domain, a C-terminal IL-1-like cytokine domain, and a central helix-turn-helix (HTH) domain. The N-terminal domain consists of a highly conserved nuclear localization sequence (NLS), which contains the chromatin binding module (CBM, AA40–56) and the core of the NLS (AA61–78). The NLS of IL-33 plays an important role in maintaining immune homeostasis, while the CBM of IL-33 binds to histones H2A and H2B and accumulates in the nucleus [13]. Upon cell necrosis, IL-33 complexed with histones is released and activated, allowing it to bind to its specific extracellular receptor ST2. In the absence of the CBM, IL-33 is rapidly released into the extracellular space, leading to a disturbance in the homeostasis of the body. The central domain of IL-33 is processed by enzymes to generate the mature form of IL-33 produced by mast cells and neutrophils. The IL-1-like cytokine domain of IL-33 is a typical β-clover fold, which is highly similar to IL-1α, IL-1β, and IL-18. This domain contains an important site for binding to ST2 expressed by target cells, and the resulting IL-33/ST2 complexes activate downstream cytokines (Figure 1C) [11,14].

### 2.2. Cleavage Sites and Biological Activity of IL-33

IL-33 is secreted in a precursor form and performs the dual functions of transcriptional regulation and mediating pro-inflammatory effects. In the resting state, the IL-33 precursor enters the nucleus under the guidance of the helix-turn-helix-like domain at the N-terminal site and binds to specific parts of the chromosome to regulate gene expression. In the activated state, induced by conditions such as mechanical stress, injury, or necrosis, the IL-33 precursor is hydrolyzed by the corresponding protease into an activated form and released from the cell to trigger an inflammatory response. It was initially thought that the IL-33 precursor (30 kDa) was not biologically active and required caspase-1 cleavage to form the mature fragment (18 kDa) in order to activate the ST2 receptor [15]. However, in vitro experiments using THP-1 cells showed that caspase-1 cleaved only the IL-1β precursor and not the IL-33 precursor directly [16]. Other studies also showed that the IL-33 precursor is different from IL-18 (IL-1F4) and that IL-33 maturation does not require the participation of caspase-1. Conversely, caspase-1 hydrolysis substantially reduced IL-33 levels and attenuated the pro-inflammatory properties of IL-33 [17,18].

IL-33 precursors can be proteolytically hydrolyzed by apoptosis-related caspases [19,20]. For example, caspase-3 and caspase-7 can function at human DGVD178 and mouse DGVD175 sites, while caspase-1, caspase-4, and caspase-5 cannot cleave the IL-33 precursor but are capable of cleaving the IL-1β precursor. However, activation of the IL-33 precursor does not depend on the hydrolysis of caspase series proteins and is only hydrolyzed by caspase-3 and caspase-7 during apoptosis, resulting in the loss of its pro-inflammatory activity [21,22]. Other studies have shown that full-length IL-33 can still be translocated into the nucleus after cleavage by caspase-3 at the AA175 site to regulate gene transcription [23]. In addition, the activation and secretion of Pro-IL-33 depend on cleavage by calpain, which is activated when the intracellular calcium concentration increases. The resulting active forms of IL-133 (24 kDa and 23 kDa) are then secreted extracellularly. This process is similar to the caspase-mediated cleavage of Pro-IL-1a to generate the mature form for extracellular secretion [24].

Neutrophil proteinase 3 (PR3) is currently recognized as an IL-1 family activating enzyme, which functions by cleaving IL-1 family precursors, such as IL-1 and IL-18 precursors, into their activated forms. PR3 processing of the IL-33 precursor has dual functions of activation and termination mediated by and acting on different sites to produce different forms of IL-33. Human IL-33/p1 is the highly active form, while human IL-33/p2 and IL-33/p3 are inactive forms. Furthermore, the activation of IL-33 precursors is time dependent, in that the cleavage activity of PR3 first increases with stimulation time and then decreases [25].

Furthermore, Lefrancais et al. showed that the precursor form of IL-33, IL-33_1-270_, had intrinsic biological activity [26]. IL-33_1-270_ derived from three different sources, rabbit reticulocyte lysate (RRL), wheat germ extract (WGE), and human, stimulated MC/9 mast cells, to produce interleukin 6 (IL-6). IL-33_1-270_ is cleaved by neutrophil serine proteases cathepsin G and elastase to deliver more biologically active forms, including IL-33_95-270_, IL-33_99-270_, and IL-33_109-270_ (Figure 1B) [27].

### 2.3. ST2 Pathways under Physiological Conditions

#### 2.3.1. Molecular Structure and Function of ST2

For many years, suppression of tumorigenicity 2 (ST2), also known as IL1RL1, DER4, T1, and FIT-1, was considered to be an “orphan” receptor lacking a specific ligand until its specific interaction with IL-33 was demonstrated [28]. Then, in 1989, Tominaga et al. showed that the ST2 gene in the BALB/c-3T3 cell line has a similar structure to other IL-1 receptors (IL-1R), including the Toll/IL-1R domain consisting of a central five-stranded β-sheet surrounded by five α-helices located at the cytosolic end of the protein [29]. The human ST2 gene, spanning approximately 40 kb on human chromosome 2q12.1, is transcribed and translated into a protein containing 556 AA with a molecular weight of approximately 63 kD [30].

Of the four subtypes of ST2 generated by alternative splicing, ST2L is a membrane-anchored receptor and is highly homologous to the IL-1 type 1 receptor, with extracellular, transmembrane, and cytoplasmic domains [31,32]. Intriguingly, ST2L is widely expressed on the surface of various cells, including Th2 lymphocytes, dendritic cells, macrophages, fibroblasts, mast cells, and eosinophils [33,34]. Soluble ST2 (sST2) is a secreted isoform with an extracellular domain identical to ST2L but with an additional nine and five AAs in mice and humans, respectively [28,35]. The other two mRNA isoforms encode ST2V, which is similar to sST2 but lacks a third extracellular immunoglobulin domain, and a hydrophobic tail generated by alternative splicing of the C-terminal portion of ST2. Characteristically, these subtypes are predominantly expressed in the gut, and overexpression in cell lines results in restricted membrane localization [36]. Lastly, ST2VL is generated by alternative splicing, which results in the deletion of the IL-1RL1-b transmembrane domain. Although ST2LV has been described in Gallus gallus, no equivalent has been reported in humans [37].

#### 2.3.2. IL-33/ST2 Signaling under Physiological Conditions

The cytokine activity of the alarmin IL-33 is dependent on the specific receptor ST2. Following the binding of IL-33 to ST2, its accessory receptor IL-1RAcP is recruited to form the functional IL-33 receptor complex (IL-33R1) [38]. Once IL-33R1 binds to IL-33, the receptor complex is activated by exposure of the TIR domain of IL-33R1 and recruitment of a TIR-domain-containing signaling adapter (i.e., myeloid differentiation primary response gene 88 (MyD88) or MyD88 linker-like 1 Signal transduction of the complex (MAL1)) into the heterodimeric IL-33R1 complex. Recruitment of MyD88 or MAL1 results in the recruitment of IL-1 receptor-associated kinase 1 (IRAK1) and IRAK4 via their death domains. IRAK4 in turn activates IRAK1 and promotes IRAK1 autophosphorylation [9,39]. This complex then activates downstream signaling molecules, including mitogen-activated protein kinases MAPK and NF-κB [40]. Human IL-33 can also activate the phosphoinositide-3 kinase (PI3K) and mTOR pathways in immune cells, including Th2, eosinophils, and macrophages [41]. These downstream signaling events induce the expression of several pro-inflammatory cytokines, depending on the target cell. For example, IL-33 induces the secretion of IL-5 and IL-13 by Th2 cells [42,43]. Mast cells treated with IL-33 produce IL-4, IL-5, and IL-6 [44], while primary keratinocytes exposed secrete IL-6 and TNF-α [45].

## 3. Immune Cell Targets of IL-33 in the Development of COVID-19 Pathology

The immune response disorder caused by SARS-CoV-2 infection is one of the main causes of multiple-organ damage and death associated with COVID-19. SARS-CoV-2 invades almost all types of immune cells, leading to dysregulation of innate and adaptive immune responses, hypolymphemia, and excessive release of inflammatory factors, including IL-33. Liang et al. revealed that a relatively high abundance of IL-33 in epithelial and endothelial cells was positively correlated with adverse pathological outcomes in COVID-19 patients. This phenomenon may be due to the epithelial damage caused by the strong interaction between airway epithelial cells and activated immune cells. However, the mechanism by which IL-33 regulates the pathological development of COVID-19 and its potential ability to target cells of the innate and adaptive immune systems have not yet been extensively explored.

### 3.1. Neutrophils

Neutrophils are the most abundant white blood cells in the human immune system, accounting for between 50% and 70% of this population. Following an attack by pathogenic micro-organisms, including bacteria, fungi, and viruses, neutrophils kill and remove pathogens and damage tissues through activation, chemotaxis, infiltration, phagocytosis, and degranulation. Neutrophils also secrete antimicrobial peptides, reactive oxygen species (ROS), and pro-inflammatory mediators that regulate inflammatory responses [46,47]. Previous studies have shown that accumulated neutrophils release an extracellular network of enucleated chromatin to trap and immobilize pathogens, thereby preventing the spread of infection [48]. This mechanism was first reported by Brinkmann et al. in 2004, and the structures were named neutrophil extracellular traps (NETs) [49], which consist of a network structure, with DNA as the skeleton, inlaid with histones, myeloperoxidase, neutrophil elastase, cathepsin G, protease 3, and other proteins with bactericidal and permeability-increasing effects. NETs function as a double-edged sword, playing anti- or pro-inflammatory roles in immune responses, including COVID-19. Previous studies have shown elevated concentrations of NETs in the blood and lung tissues of patients who are critically ill with COVID-19. Moreover, serum from patients with COVID-19 has been shown to directly induce NET production by healthy neutrophils in vitro [50]. Excessive release of NETs triggers an inflammatory cascade, which destroys the surrounding tissues, promotes micro-thrombosis, and causes permanent damage to organs such as the lung, kidney, and cardiovascular system, suggesting that the release of NETs plays a key role in the pathological changes seen in COVID-19 patients [48,51,52], although the underlying mechanism remains to be explored.

Both in vivo and in vitro studies have shown that the release of NETs is highly dependent on inflammatory factors, such as IL-33, TNF-α, IL-1β, and ROS [53]. IL-33, an important cytokine of the IL-1 family, is released from apoptotic or necrotic cells and targets various immune cells, including macrophages and neutrophils, by specifically binding to its orphan receptor ST2. In sepsis, IL-33 promotes rapid neutrophil infiltration, migration, and activation by enhancing macrophage-derived CXCL1 and CXCL2 expression [54]. IL-33 released from hepatic sinusoidal endothelial cells stimulates neutrophils to form NETs after liver ischemia–reperfusion injury [55]. A similarly large infiltration of neutrophils is observed in the lungs of COVID-19 patients, suggesting that IL-33 also induces NET formation in the lungs in response to SARS-CoV-2 infection. Knockdown of the ST2 gene in mouse bone-marrow-derived neutrophils significantly reduced production of the NET marker MPO–DNA complexes in the culture medium, and this effect was not reversed by treatment with recombinant IL-33, indicating that the IL-33/ST2 signaling pathway is directly involved in the production of NETs by neutrophils. Furthermore, in a model of gout as the most common form of inflammatory arthritis, IL-33 was shown to promote neutrophil influx and trigger neutrophil-dependent ROS production via ST2, which activates the transient receptor potential ankyrin 1 channel in the dorsal root ganglion (DRG) neurons and induces nociception [56]. However, ROS production is an essential step in the production of NETs by neutrophils. Considering that IL-33/ST2 axis deficiency exacerbates neutrophil-dominant allergic airway inflammation, which is mainly induced by NETs, it is reasonable to speculate that the IL-33/ST2 signaling pathway plays an important role in the generation of NETs.

In conclusion, we speculate that IL-33 is likely to exacerbate tissue and organ damage by enhancing neutrophil migration, infiltration, and aggregation in the development of COVID-19 pathology. Moreover, IL-33 induces massive ROS accumulation in neutrophils, leading to increased release of NETs dependent on the ST2 pathway. Furthermore, neutrophil elastase and cathepsin G in NETs can also promote IL-33 processing and maturation, thus forming a positive feedback regulation loop between IL-33 and NETs, triggering an inflammatory cascade reaction, and forming a vicious cycle that causes irreversible damage to multiple organs. Although the exact details of the interaction between IL-33 and neutrophils during the progression of COVID-19 are yet to be fully elucidated, we propose that the IL-33/ST2/ROS/NET pathway is an important basis for further in-depth studies that will provide a comprehensive understanding of the pathogenesis of COVID-19 and highlight the checkpoint molecules on this pathway that may yield novel strategies for the prevention and treatment of patients with COVID-19 (Figure 2).

### 3.2. Innate Lymphoid Type-2 Cells

Innate lymphoid cells (ILCs) are a newly discovered family of cells with acquired immune function. ILCs are tissue-resident lymphocytes distributed predominantly in the mucosal barrier and perform early immune surveillance and immune regulation by secreting cytokines [57]. ILCs also participate in the formation of mucosal immunity and play an important role in the development of lymphocytes, the repair of tissue damage, and the maintenance of the epithelial barrier [58]. Based on the expression of specific transcription factors and cytokines, ILCs can be divided into three subgroups: ILC1, ILC2, and ILC3 [59,60]. ILC2 is a class of innate non-B/non-T lymphocytes, mainly derived from lymphoid progenitor cells, regulated by the nuclear transcription factor retinoic acid-related orphan receptor (related orphan receptor α, RORα) and GATA3 [61,62]. ILC2s function as a bridge between the innate and adaptive immune systems by producing type 2 cytokines, which mediate type 2 immune responses and play an important role in the Th1/Th2 balance. ILC2 plays an important role in allergic diseases [63,64], inflammatory responses [65,66], acute respiratory distress syndrome (ARDS) [67], and COVID-19 [68]. Increased levels of the checkpoint cytokines IL-33 and ILC2 frequencies were detected in the serum and plasma of patients with moderate COVID-19 [69], while decreased ILC2 frequencies were observed in severe patients [70,71], indicating that ILC2 may be involved in COVID-19 pathology. Activation of ILC2 inhibits IFN-γ and natural killer (NK) cell-mediated innate anti-tumor immunity [72], suggesting that ILC2 also has a role in suppressing innate immunity in COVID-19 patients. Previous studies have shown that ILC2s are stimulated by the alarmins IL-33 and IL-25 to produce large amounts of type 2 cytokines IL-13 and IL-5 and mediate both innate and adaptive type 2 immune responses [42]. In addition, IL-13 is essential for maintaining mucosal homeostasis, and overexpression of IL-13 increases airway hyperresponsiveness, exacerbates hypersensitivity pneumonitis, and limits airway tissue remodeling [73]. Based on the basic pathological features of COVID-19, which include dysregulation of the lung mucosa and the massive production of mucus, it is reasonable to speculate that IL-13 is likely to be closely related to the pathological progression to severe COVID-19. Treatment with the IL-13/IL-4 signaling antagonist, dupilumab, significantly reduced pathological damage in COVID-19 patients. Dupilumab treatment also reduced mortality in mice by attenuating SARS-CoV-2-induced pathological damage, suggesting that IL-13 has a role in exacerbating post-COVID-19 immune damage [74]. Interestingly, IL-13 is thought to decrease the expression of angiotensin II (ACE2) and increase the expression of the host protease transmembrane serine protease 2 (TMPRSS2) in bronchial epithelial cells in vitro [75]. SARS-CoV-2 gains entry into lung type II epithelial cells by binding ACE2 in co-operation with TMPRSS2. We propose that ILC2-derived IL-13 mediates increased TMPRSS2 expression to counteract the protective effect of decreased ACE2 expression, which may be related to the peptidase-dependent protective effects of ACE2 in viral infection-induced acute lung injury [76]. In addition, IL-13 upregulates the downstream hyaluronan synthase 1 (Has1) gene and promotes the deposition of hyaluronic acid (HA) polysaccharide in the lung parenchyma, which may be a new mechanism for the development of severe COVID-19. Although ILC2/IL-13 plays an important role in the pathological changes of COVID-19, the specific regulatory mechanism remains to be further explored.

As a key upstream cytokine of ILC2/IL-13, IL-33 activates ILC2s in response to various stimuli. IL-33 induces the production of type 2 cytokines, such as IL-5 and IL-13, and promotes the production of eosinophils or goblet cell proliferation, which plays an important role in tissue damage repair and allergic responses [77,78]. When alveolar epithelial cells are infected by SAR-CoV-2, IL-33 derived from damaged tissue or alveolar macrophages and NKT cells activate NF-κB and MAPK via the IL-33/ST2 pathway, increasing the proliferation and activity of ILC2 [79]. Specifically, IL-33 mediates the phosphorylation of the p38-activated transcription factor GATA3 through MyD88 activation induced by ST2/IL1RAcP dimerization. The phosphorylated form of GATA3 then binds to the IL5 and IL13 promoters in ILC2 to upregulate the expression of these cytokines. IL-33 also induces the production of IL-6, but in a GATA3-independent manner, thereby exacerbating the injury caused by the pulmonary cytokine storm [80]. Interestingly, IL-33 also induces non-canonical phosphorylation of STAT3 at S727 through MAPK, which promotes the transport of pSTAT3 into the mitochondria, enhances cellular respiration and ATP generation, and further promotes the methionine cycle in ILC2 mitochondria. The accumulation of methyl donors increases the abundance of the histone modification H3K4me3 in the IL5 and IL13 loci, which in turn upregulates the expression of the encoded cytokines [81]. The ability of IL-33 to activate both the IL-33/p38/GATA3 and IL-33/MAPK/STAT3 pathways suggests that IL-33 plays a key role in the regulation of ILC2 in pulmonary inflammation, which provides a new perspective for research into the mechanism of the secondary pulmonary symptoms of COVID-19. However, whether the IL-33/MAPK/STAT3/IL-13 axis plays a positive or negative role in the pathology of COVID-19 requires further exploration (Figure 3).

### 3.3. Dendritic Cells

Dendritic cells (DCs), as efficient professional antigen-presenting cells responsible for bridging the gap between innate and adaptive immunity, are essential in recognizing pathogens and secreting inflammatory mediators [82,83]. DCs arise from bone marrow progenitors, known as common myeloid progenitors (CMPs). Dendritic cells can be divided into two subpopulations based on the expression of surface markers: conventional dendritic cells (cDCs) and plasmacytoid dendritic cells (pDCs) [84,85,86]. cDCs are primarily responsible for presenting antigens and stimulating naive T-cell expansion and differentiation. pDCs typically produce high amounts of type I IFN (IFN-I) and pro-inflammatory cytokines, including IL-6 and TNF-α, in response to in vivo RNA and DNA viral infection [87,88,89,90]. IL-6, IL-1β, TNF-α, and IFN-I are released by pDCs, which are key factors in cytokine storm and correlate with disease severity. According to current authoritative journal reports, the cytokine storm generated by the activation of various inflammatory signaling pathways is the hallmark pathological change of COVID-19 [91,92]. Following a pathogen attack, activation of immune cells (T cells, dendritic cells, macrophages, natural killer cells, and cytotoxic lymphocytes) occurs, resulting in the release of cytokines and chemokines, and subsequently leading to an inflammatory response for viral clearance. Cytokines, in turn, may act on different cells, nearby cells, distant cells, as well as on the cell that secretes them. Notably, in the initial stage, the moderate release of cytokines exhibits beneficial inflammatory effects, acting only on viral cells, but after the over-activation of the immune system, the over-produced cytokines also rapidly kill the host cells [93]. However, the interaction mechanisms between immune cells, especially pDCs and cytokines, as well as between cytokines themselves, in the pathogenesis and development of COVID-19 remain incompletely characterized.

Interferon is one of the most important antiviral cytokines whose function declines or defects can lead to reduced resistance to viral infections, including but not limited to COVID-19 [94,95,96]. Type I interferon (IFN-I) mainly includes three subtypes, IFN-α, β, and ω, which can participate in the antiviral response of immune cells through the JAK–STAT signaling pathway and the induction of interferon-stimulated genes (ISGs) [97]. Numerous studies have shown that impaired IFN-I responses are associated with severe and critical COVID-19 infections. For instance, pre-existing autoantibodies to IFN-I in COVID-19 patients with APS-1 can induce severe pneumonia [98,99]. Therefore, rapid induction of IFN-1 production in vivo can effectively control early pathological damage after COVID-19 infection [100]. Although IFN-I plays a protective role in the early stage of COVID-19 infection, the continuous increase in IFN-I levels can exacerbate abnormal inflammatory responses and aggravate pathological damage through the cGAS–STING pathway in the late stage of COVID-19 infection [101]. As is well known, IFN-I can be produced in almost all cell types, but pDCs have been recognized as the main source of IFN-I because the production of IFN-I depends on the Toll-like receptor-7 (TLR7)/TLR9 pathway, through which TLR7 and TLR9 are specifically expressed in pDCs [102]. Upon COVID-19 infection, TLR7 or TLR9 activates MyD88 and IL-1 receptor-associated kinase 4 (IRAK-4), which then interact with tumor necrosis factor receptor-associated factor-6 (TRAF6), TRAF3, IRAK1, IKKα, and interferon regulatory factor 7 (IRF7) interaction. Ultimately, IRAK-1 and IKKα phosphorylate IRF7, leading to IRF7 activation and induction of IFN-I gene transcription and massive IFN-I production [103,104]. Although the reduction in IFN-I production by pDCs is likely to be an important factor in the pathological aggravation of COVID-19, the specific molecular mechanism of their mutual regulation remains unclear.

IL-33, as a major member of the cytokine storm, plays a key role in the pathological changes after COVID-19. However, whether the interaction mechanism of IL-33 with pDC or IFN-I is involved in the pathological development of COVID-19 remains unknown. Previous studies found that IL-33 treatment significantly limited IFN-α production, at least in part, by downregulating the secretory function of pDCs [105,106]. Specifically, IL-33 affects TLR7-mediated pDC activation by rapidly depleting the intracellular adaptor molecules IRAK1 and viperin, resulting in a hyporesponsive state of TLR7. The ability of IL-33 to reduce the secretion of IFN-I by pDCs is dependent on the specific receptor ST2 localized on the surface of pDCs [105,106]. Since the TLR7 pathway is required for recognition of the SARS-CoV-2 genome and production of IFN-I, the IL-33/ST2 axis may suppress innate antiviral immunity and delay viral clearance in COVID-19 patients by reducing IFN-I expression (Figure 4). Additionally, an IFN-α/λ^low^ IL-33^high^ cytokine microenvironment allows for the onset of type 2 immune response and early viral growth [105]. This in turn results in persistent changes in alveolar epithelial cells and immune cells, which are the immune basis of severe lung injury following COVID-19 infection. Collectively, the IL-33/ST2 axis remarkably reduces pDCs-dependent IFN-I production by inhibiting the biological activity of the IRAK1/TRL7 pathway, thereby increasing the susceptibility of COVID-19 patients to the virus in the early to middle stage, especially for patients with insufficient circulating IFN levels. This may well explain the clinically milder and more self-limiting symptoms in younger patients and more severe systemic pathological reactions and complications in older patients. Therefore, targeting the intervention of IL-33/ST2/pDCs/IFN-I signaling pathway may provide new therapeutic opportunities for COVID-19.

### 3.4. Macrophages

Macrophages are widely and abundantly distributed in various tissues and organs and play a key role in organ development, homeostasis, and injury repair in a tissue-specific manner [107]. As the most important innate immune cells, macrophages are key effector cells in the middle and late stages of immune regulation of invading pathogens (such as bacteria, fungi, and viruses) [108]. Alveolar macrophages (AMs) are the first line of defense in the airways and alveolar spaces of the lungs and perform this function through exposure, identification, and removal of pathogens and particulate matter that enter the airways during respiration through phagocytosis. A large number of bone-marrow-derived leukocytes, including neutrophils, monocytes, and macrophages, were observed in lung tissue biopsies of patients with COVID-19, indicating that the pathological changes in these patients are caused by massive leukocyte proliferation, activation, and infiltration [109]. The mechanisms of macrophage proliferation and migration are complex and involve the activity of metalloproteinases, chemokines, and cytokines [110,111,112]. These factors form a positive feedback regulation loop in macrophages and trigger an inflammatory cascade reaction, forming a vicious cycle that causes irreversible damage to multiple organs. IL-33 plays a crucial role in the regulation of macrophage function. IL-33 treatment inhibited phagocytosis by AMs, resulting in decreased clearance of pathogenic micro-organisms and apoptotic cells from the airways [113]. Indeed, IL-33 not only reduces the phagocytic function of AMs but also enhances their secretory function. For instance, the IL-33/Akt1 pathway regulates pulmonary fibrosis by enhancing the production of the profibrotic cytokine IL-13 by macrophages [114]. The IL-33/STAT3 pathway exacerbates acute lung injury and acute respiratory distress syndrome by increasing AP-1, ERK1/2/CREB, and NF-κB-mediated expression of MMP2 and MMP9 [114,115]. These studies suggest the involvement of IL-33-targeted macrophages in COVID-19 pathology, although the specific mechanisms remain to be further explored.

Polarization of macrophages to different phenotypes, including M1 and M2, is an efficient mechanism of adaptation to various stimuli. M1 macrophages are mainly stimulated by IFN-γ and LPS, while M2 macrophages are mainly stimulated by Th2-type cytokines [116,117]. M2 cells can be divided into three subtypes: M2a, also known as alternately activated macrophages (AAM), induced by IL-4 or IL-13; M2b, also known as type II macrophages, induced by immune complexes and bacterial LPS; and M2c, also known as inactivated macrophages, induced by TGF-β or glucocorticoids [118]. In the resting state, AMs are usually maintained in a quiescent state similar to the M2c phenotype to avoid damage to the alveoli. In the activated state (such as SARS-CoV-2 infection), AMs may be responsible for triggering the inflammatory response [119]. Previous studies have shown that IL-33 induces macrophage polarization toward the M1 phenotype, which exacerbates airway inflammation. IL-33 can also induce pulmonary fibrosis by promoting ST2/IL-13/IL-4Rα-mediated AAM activation [120,121].

Overall, we speculate that IL-33 functions in the development of COVID-19 pathology by exacerbating tissue and organ damage by enhancing the proliferation, migration, and activation of macrophages. Moreover, IL-33 promoted AP-1, ERK1/2/CREB, and NF-κB-mediated expression of IL-13, MMP2, and MMP9 in macrophages. IL-33 also promotes the polarization of macrophages to the pro-inflammatory M1 type and forms a positive feedback regulation loop, which triggers an inflammatory cascade reaction, forming a vicious cycle that results in irreversible damage to multiple organs (Figure 3).

### 3.5. CD4^+^ T Cells

In mammals, T cells are key members of the immune system, playing a critical role in all aspects of the immune response from effective clearance of pathogens to the establishment of memory responses and rapid restoration of immune homeostasis. CD4^+^ T can recognize antigens via their T-cell-specific receptors (TCRs), leading to rapid clonal expansion, and differentiation into functionally distinct Th subpopulations, thereby coordinating the immune response at the site of inflammation. IL-33 is involved in regulating the entire process of differentiation, migration, and effector responses of CD4^+^ T cells. Initial CD4^+^ αβ T cells barely express ST2 on their surface; however, following TCR-mediated activation by antigen-presenting cells, these cells are induced to differentiate into subsets, including Th1, Th2, and Th17 [122]. Of these, Th2 cells were first confirmed to express ST2 receptors on their surface, and their activation is closely associated with TCR/STAT5 signaling [123]. Thus IL-33 acts directly or indirectly as a chemoattractant for CD4^+^ αβ T cells, promoting CD4^+^ ST2^+^ Th2 cell responses and upregulating the expression of factors such as IL-4, IL-5, and IL-13 [124,125]. It is also interesting to note that during lymphocytic choroid plexus meningitis virus (LCMV) infection, IL-33 also transiently upregulates ST2 receptor expression by Th1 cells through expression of the transcription factor T-bet and IL-12-dependent STAT4 signaling [126], suggesting that IL-33 is also involved in Th1 cell responses. Not surprisingly, IL-33 may play both a deleterious and protective role in COVID-19 by regulating CD4^+^ T cells. On the one hand, IL-33 induces the ability of CD4^+^ T cells to differentiate into a range of helper and effector cell types in SARS-CoV-2 infection, participating in directing antibody secretion by B cells, helping CD8^+^ T cells, and recruiting innate cells, as well as its direct antiviral activity and promoting tissue repair [127]. On the other hand, hyper-elevation of type 2 cytokines promotes differentiation of pathogenic γδ T cells (IFN-γ^low^ GM-CSF^high^), which in turn causes severe respiratory immune dysregulation. Although anti-IL-33 treatment has been shown to attenuate deleterious type 2 memory responses during rhinovirus infection, thereby reducing airway hyperresponsiveness [128], the specific mechanisms by which IL-33 regulates CD4^+^ T cells in SARS-CoV-2 infection remain to be explored.

### 3.6. Th17/Treg Cells

It is well known that naive CD4^+^ T cells can differentiate into several effector cell subsets, which dictate their function and the cytokines they release. These include type 1 helper T (Th1) cells, type 2 helper T (Th2) cells, type 17 helper T (Th17) cells, and regulatory T (Treg) cells. Th17 lymphocytes are induced by retinoic acid-related orphan receptor (ROR) γt, RORα, and STAT3, and secrete IL-17A, IL-17F, IL-21, IL-22, and CCL20. As such, Th17 cells are thought to be essential for autoimmune inflammation. Treg cells, on the other hand, are induced by the forkhead box protein P3 (Foxp3) and control excessive immune responses by secreting suppressive cytokines, such as TGF-β and IL-10, or through cell-mediated immune checkpoint inhibitors, such as TIGIT and CTLA-4 [129]. Studies have demonstrated that Th17/Treg homeostasis plays an important role in the severity of lung injury and the course of the systemic inflammatory response characterized by acute lung injury and ARDS [130,131]. A growing body of literature suggests that IL-17 and GM-CSF levels are elevated in the peripheral blood of patients with COVID-19 and that the proportion of Th17 cells in the bronchoalveolar lavage fluid is significantly elevated in these patients [132]. We suggest that the elevated Th17 cells are likely to be important in the development of severe immune damage associated with COVID-19 pneumonia through the release of pro-inflammatory factors, such as IL-17, GM-CSF, IL-21, and IL-22, which promote neutrophil migration while downregulating Treg cell expression [133]. On the other hand, a recent study of the Tregs (CD4^+^ FoxP3^+^ CD25^+^) in PBMC of COVID-19 patients treated in intensive care units showed a significant decrease in Treg numbers and a similar decrease in the expression of FoxP3 mRNA and immunosuppressive cytokines (IL-10 and TGFβ) [134]. These findings suggested that in severe COVID-19 cases, excessive inflammation and tissue damage may be further exacerbated by a significant reduction in the Treg population, which attenuates or suppresses overactive innate immune responses. Clinically, it was noted that the Th17/Treg cell expression ratio, the IL-17/IL-10 cytokines ratio, and the RORγt/Foxp3 transcription factor ratio were significantly higher in deceased patients than those in recovered patients. These findings indicate that targeting the dysregulation of the Th17/Treg cell population ratio may be a new strategy for the treatment of excessive immune injury in COVID-19 pneumonia.

In the Th cell population, IL-33 is primarily associated with Th2-associated immune responses [135]. However, a growing number of studies have found that IL-33 also plays a critical role in mediating the differentiation of Th17/Treg cells through a mechanism, which may be related to IL-33-induced activation and maturation of dendritic cells (DCs), which are the major antigen-presenting cell type responsible for activation of naïve T cells. On the one hand, IL-33 upregulates MHC-II class and co-stimulatory molecules, such as CD40 and CD80, in a dose-dependent manner to induce maturation and activation of immature DCs. In turn, IL-33-matured DCs (IL-33-matDCs) further stimulate the activation of CD4^+^ T cells and induce their differentiation to Th17 cells through secretion of IL-1β and IL-6 [136]. On the other hand, IL-33-matDCs have been shown to significantly inhibit Treg differentiation while promoting the differentiation of Tregs to Th17 cells through binding of IL-6 secreted by IL-33-matDCs binds to IL-6 receptors on CD25hiTreg cells [137]. Thus, IL-33 is critical as an upstream pro-inflammatory factor that promotes the Th17/Treg imbalance in autoimmune responses. IL-33-deficiency has been reported to protect mice from dextran sulfate sodium (DSS)-induced experimental colitis by inhibiting Th17 cell responses [138], and the IL-33/ST2 axis controls Th17 immune responses that exacerbate allergic airway disease, thus implicating IL-33 as an important therapeutic target for autoimmune diseases [139]. Although the relevance of IL-33 in the Th17/Treg cell imbalance associated with COVID-19 has not been reported, we suggest that IL-33 is likely to similarly induce an imbalance in the Th cell subsets locally in the lung and also systemically in a DC-dependent manner, resulting in irreversible hyperimmune damage and inflammatory responses. Thus, it can be speculated that blockade of the IL-33/Treg/Th17 signaling axis or the IL-6 receptor may then hold promise for the development of novel therapeutic strategies for COVID-19 (Figure 2).

### 3.7. CD8^+^ T Cells

It is widely accepted that CD8^+^ T cells are essential for the clearance of many viral infections, including SARS-CoV-2, through their ability to kill infected cells. In COVID-19 patients, the CD8^+^ T cell population undergoes quantitative and qualitative changes [140], with decreased cell numbers and activation phenotypes frequently observed, particularly in severe disease [141,142,143,144,145]. Early studies revealed that CD8^+^ T cells are specific for a range of SARS-CoV-2 antigens [146], including spike, nucleocapsid, and membranous proteins, as well as other non-structural proteins. The presence of these specific CD8^+^ T cell responses to SARS-CoV-2 is predictive of better outcomes in COVID-19 patients [147,148]. Limited viral clearance in the respiratory tract was observed in CD8^+^-depleted convalescent rhesus macaques upon SARS-CoV-2 rechallenge, implying that memory CD8^+^ T cells are required for SARS-CoV-2 clearance [149]. However, the mechanism by which CD8^+^ T cells are regulated in COVID-19 and whether they are depleted remains an unsolved mystery.

CD8^+^ T-cell effector functions are influenced by the inflammatory micro-environment in the tissues, including different cytokines. Early research showed that IL-33 is required for cytotoxic CD8^+^ T-cell responses and antiviral immune responses to viral infection in mice lacking IL-33 or its receptor and that IL-33 is irreplaceable for CD8^+^ T-cell expansion, production of multiple cytokines, and acquisition of cytotoxic function [150]. Yang et al. showed that ST2 expression in Tc1 polarized conditions is regulated by T-bet, a major Th1/Tc1 transcription factor, whereas CD8^+^ T cells cultured under Tc2 conditions do not express ST2 [151]. In addition, Gadd45b mediates IL-33 in concert with IL-12 to stimulate IFN-γ production in Tc1 cells and promotes CD8^+^ T-cell effector functions. In conclusion, IL-33 is essential for the regulation of CD8^+^ T cells. Thus, IL-33 may induce T-bet-dependent differentiation of CD8^+^ T cells to form cytotoxic as well as memory CD8^+^ T cells in COVID-19 and upregulate CD8^+^ T-cell expression of GM-CSF, which may be responsible for neutrophil and monocyte activation in the tissues [152]. Given that IL-33 is critical for the function of CD8^+^ T cells in the antiviral response to persistent SARS-CoV2 infection and may contribute to viral elimination and that IL-33-induced hyperinflammation conversely exacerbates COVID-19-related complications, a better understanding of the exact role of IL-33 and how the IL-33/ST2 axis is manipulated is of vital importance to the development of effective strategies for the prevention and treatment of COVID-19.

## 4. Conclusions

In 2020, COVID-19 emerged as a major public health challenge. Serious complications and sequelae, including acute lung injury, acute respiratory distress syndrome, systemic coagulation dysfunction, and idiopathic pulmonary fibrosis, still cannot be effectively prevented and treated. Although immune cell deficiency and inflammatory cytokine storm are the hallmark pathological mechanisms of COVID-19, the exact mechanisms are still poorly understood. IL-33 released by airway epithelial cells functions as an important checkpoint cytokine, which plays an important role in immune cell regulation and cytokine regulation in COVID-19. In the acute phase, IL-33 promotes neutrophil proliferation, migration, and invasion via the IL-33/ST2 signaling pathway, thereby enhancing oxidative stress, inducing marked accumulation of NETs, and ultimately causing acute damage to the heart, lung, and kidney systems. Furthermore, IL-33 binds to the ST2 receptor on the surface of pDCs and inhibits the secretion of IFN-I, greatly reducing the ability to limit virus propagation. In the subacute phase, IL-33 induces an imbalance in the ratio of Th17/Treg cell populations locally in the lungs, and even systemically, in a DC-dependent manner, resulting in inflammatory cytokine storms and irreversible immune damage. In the middle and late stages of the disease, IL-33 induces ILC2 hyperactivation, differentiation of alternately activated M2 macrophages, and the secretion of TGF-β and IL-13 from mast cells, resulting in the transformation of epithelial cells to mesenchymal cells and leading to pulmonary fibrosis. Despite some understanding of the biological properties of IL-33 and the molecular mechanism underlying its effects on target cells in COVID-19, further studies are required to confirm the functional properties of IL-33, such as the cleavage site, the structure of the mature form, its activity, and immune cell. In conclusion, inflammatory cytokine storm factors represented by IL-33 are of great significance in the monitoring and prognosis of COVID-19 to improve the early detection of patients who are severely affected by the infection and guidance for clinical treatment.

## Figures and Tables

**Figure 1 ijms-23-13656-f001:**
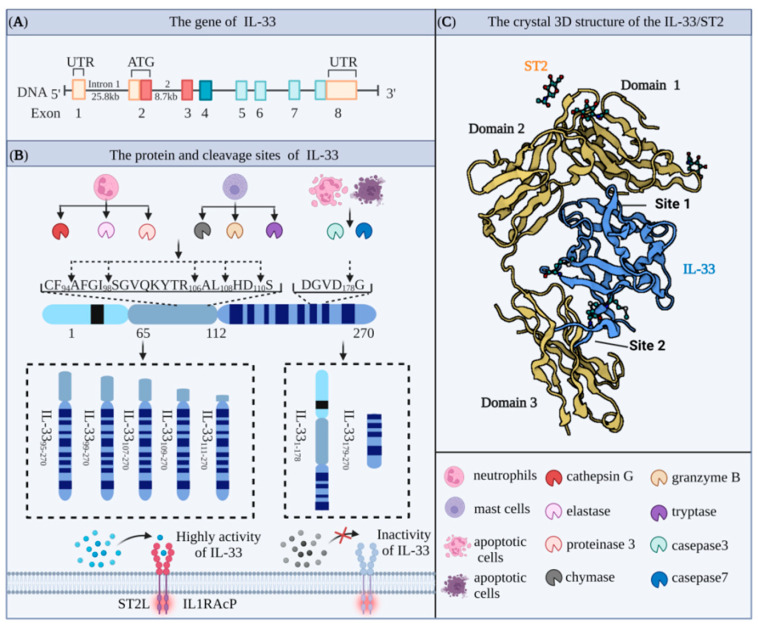
Molecular biological characteristics of IL-33 (**A**) The gene spans of human IL-33 (approximately 48 kb) contain eight exons, which are transcribed and translated into proteins containing 270 amino acids. (**B**) The full length of IL-33 is sheared to obtain different levels of active fragments. Cleaved by cathepsin, elastase, and proteinase released from neutrophils or chymase, Genzyme B, and tryptase released from mast cells, the highly active fragments of IL-33 (IL-33_95-270_, IL-33_99-270_, IL-33_107-270_, IL-33_109-270_, IL-33_111-270_) can bind to the ST2L/IL-1RAcP dimer and activate downstream signaling pathways. In contrast, when the full length of IL-33 is cleaved by casepase3/7 released by apoptotic cells, it forms a fragment (IL-33_179-270_) without biological activity. (**C**) The 3D structure of the IL-33/ST2 complex (Protein Data Bank ID:4KC3). IL-33 mediates cytokine activity through the structural domain with three Ig-like structural domains in the extracellular domain of ST2 on target cells. At Site 1, the acidic residues Glu144, Glu148, Asp149, and Asp244 of IL-33 interact with the ST2 basic residues Arg38, Lys22, Arg198, and Arg35, respectively, via a critical salt bridge. At Site 2, the acidic residue Glu165 of IL-33 interacts with Arg313 of ST2 via a salt bridge, and Tyr163 and Leu182 of IL-33 form significant hydrophobic structures with Leu246, Leu306, and Leu311 of ST2.

**Figure 2 ijms-23-13656-f002:**
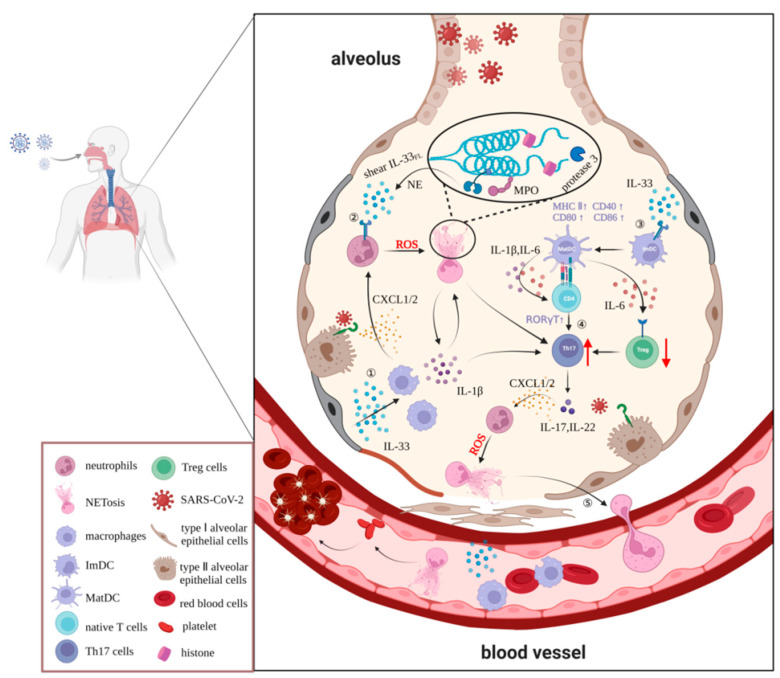
The effects and putative mechanisms of the involvement of IL-33 in neutrophils and T helper cells in COVID-19. In response to SAS-CoV-2, IL-33 is released as an alarmin from epithelial cells due to disruption of the epithelial barrier and cell damage. This promotes rapid neutrophil infiltration, migration, and activation via macrophage-derived CXCL1 and CXCL2 while inducing massive accumulation of ROS via the IL-33/ST2 signaling pathway, which in turn induces the excessive release of NETs. Additionally, IL-33 dose dependently upregulates MHC-II class and co-stimulatory molecules to induce maturation and activation of immature DCs and then stimulates activation of CD4^+^ T cells and induces their differentiation to Th17 cells instead of Treg cells through secretion of IL-1β and IL-6. Finally, a large number of NETs and inflammatory cells enter the bloodstream, triggering serious complications, such as vasoembolic conditions and inflammatory reactions.

**Figure 3 ijms-23-13656-f003:**
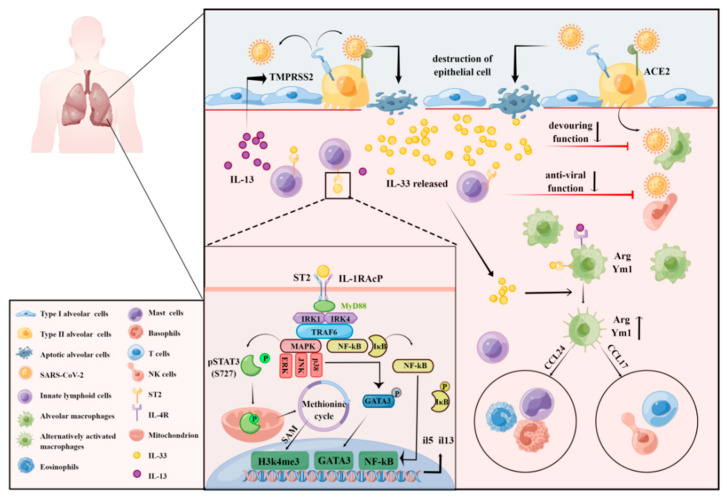
Potential mechanisms by which IL-33 regulates ILC2 and macrophages in COVID-19. SAS-CoV-2 infects cells via ACE2 on the surface of type II alveolar epithelial cells with the assistance of TMPRSS2. IL-33 acts as an alarmin in response to viral infection and is secreted in large quantities by apoptotic alveolar epithelial cells. The released IL-33 acts on the specific receptor of ST2L on the surface of the ILC2 membrane and recruits its accessory receptor IL-1RAcP to form the functional IL-33 receptor complex. Following binding of IL-33 to IL-1RAcP, the receptor complex is activated by exposure of the TIR domain of IL-1RAcP and recruitment of myeloid differentiation primary response gene 88 (MyD88) into the heterodimeric IL-33R1 complex. Recruitment of MyD88 results in the recruitment of IL-1 receptor-associated kinase 1 (IRAK1) and IRAK4 through their death domains. This complex then activates downstream signaling pathways, including mitogen-activated protein kinases MAPK (such as ERK, JNK, p38), GATA3, and NF-κB, and enhances cellular respiration and ATP production, which induces an elevated level of the histone modification H3K4me3. Activation of NF-κB, GATA3, and H3K4me3 in the nuclear membrane further promotes the production of inflammatory cytokines, such as IL-5 and IL-13, and enhances viral recruitment by TMPRSS2. On the other hand, the release of IL-33 directly reduces phagocytosis by macrophages and the antiviral effects of NK cells. In contrast, IL-33 and IL-13 bind to ST2L and IL-4R, respectively, on the surface of macrophage membranes, thereby promoting the activation of macrophage differentiation toward AAMs and damage to alveolar tissue.

**Figure 4 ijms-23-13656-f004:**
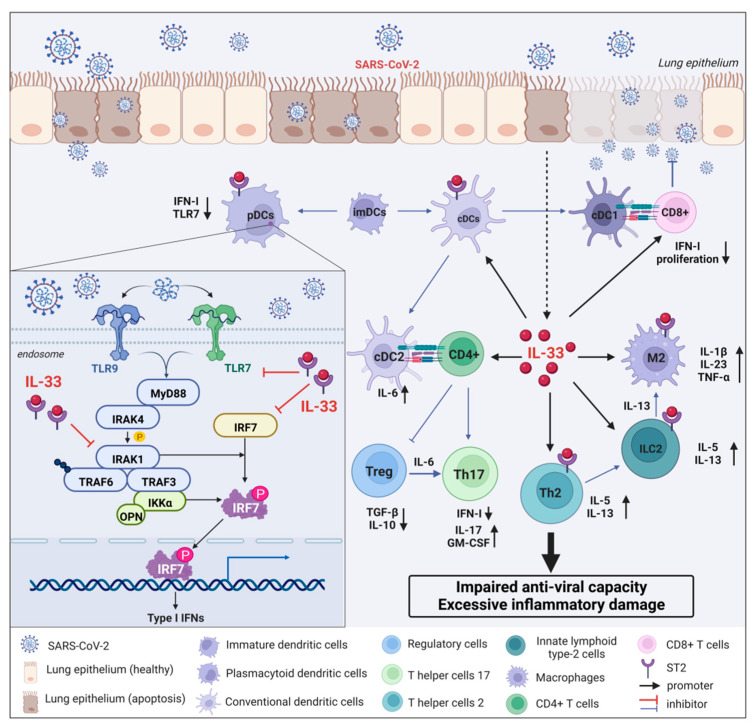
Interaction mechanism between IL-33 and IFN-I released by pDCs during COVID-19 infection. After the lung epithelium is infected with SARS-CoV-2, the replicating virus can cause epithelial cell apoptosis and directly damage the epithelium. Dendritic cells present antigens to helper T cells. Immature DCs differentiate into conventional DCs (cDCs) and plasmacytoid DCs (pDCs). IFN-I specificity derived from pDCs is dependent on the Toll-like receptor-7 (TLR7)/TLR9 pathway. Upon COVID-19 infection, TLR7 or TLR9 activates MyD88 and IL-1 receptor-associated kinase 4 (IRAK-4), which then interact with tumor necrosis factor receptor-associated factor-6 (TRAF6), TRAF3, IRAK1, IKKα, and interferon regulatory factor 7 (IRF7) interaction. Ultimately, IRAK-1 and IKKα phosphorylate IRF7, leading to IRF7 activation and induction of IFN-I gene transcription and massive IFN-I production. Additionally, IL-33, an important inflammatory storm cytokine, is abundantly released from apoptotic epithelial cells and inhibits pDC-dependent IFN-I by rapidly depleting the intracellular adaptor molecules IRAK1 and viperin, resulting in a hyporesponsive state of TLR7. Meanwhile, IL-33 induces the expression of a large number of cytokines by interacting with immune cells, such as macrophages, ILC2 cells, Th2 cells, Th17 cells, and Treg cells, which ultimately leads to abnormal inflammatory damage and decreased antiviral capacity in COVID-19 patients.

**Table 1 ijms-23-13656-t001:** Molecules involved in the cytokine storm of COVID-19.

Related Molecules	The Levels in COVID-19	Cellular Sources	Target Cells	Receptors	Drugs or Targets
IL-33	↑↑ (moderately elevated)	Endothelial cells, epithelial cells, CD45^+^ hematopoietic cells (such as macrophages and masts cells), fibroblast-like cells, myofibroblasts, glial cells, astrocytes, and oligodendrocytes.	Mast cells, type 2 innate lymphoid cells, alternatively activated macrophages, dendritic cells, Treg cells, Th17 cells, Th1 cells, Th2 cells, eosinophils, basophils, neutrophils, NK cells, CD8^+^ T cells, and iNKT cells.	ST2	IL-33 monoclonal antibody (MEDI-3506), ST2 antibody (Astegolimab), small-molecule ST2 inhibitors (under research)
IL-17	↑↑	Th17 cells, Tc17 cells, and innate immune cell populations (γδ-T cells, NKT cells, type 3 innate lymphoid cells, myeloid cells).	Neutrophils, monocytes, macrophages, mast cells, endothelial cells, fibroblasts, and type 2 innate lymphoid cells.	IL-17R	IL-17 monoclonal antibody (Bimekizumab), IL-17 receptor blocker (Brodalumab)
IL-6	↑↑↑ (significantly elevated)	T cells, B cells, monocytes, fibroblasts, kératinocytes, endothelial cells, astrocytes, bone marrow stroma cells, and mesangial cells, tumor cells (cardiac myxoma cells, myeloma cells).	T cells, B cells, endothelial cells, hepatocytes, PC12 cells, osteoclasts, mesangial cells, macrophages, keratinocytes, and oligodendrocytes.	IL-6R	IL-6 monoclonal antibody (Clazakizumab), IL-6 receptor blocker (Tocilizumab, Sarilumab)
IL-13	↑ (slightly elevated)	Type 2 innate lymphoid cells, Th2 cells, eosinophils, basophils, mast cells, and NKT cells.	Keratinocytes, Th2 cells, eosinophils, fibroblasts, and alternatively activated macrophages.	IL-4Rα	IL-13 monoclonal antibody (Lebrikizumab), IL-4Rα blocker (Dupilumab)
GM-CSF	↑↑	Epithelial cells, fibroblasts, NK cells, type 2 and 3 innate lymphoid cells, B cells, and T cells.	Monocytes, macrophages, dendritic cells, neutrophils, and eosinophils.	GMR	Anti-GM-CSF (Mavrilimumab, Gimsilumab, Lenzilumab, Otilimab)
IFN-I	↑↑	Plasmacytoid dendritic cells, T cells, B cells, NK cells, NKT cells, and APCs.	All nucleated cells	IFNAR	IFNα-2b (a recombinant interferon alpha-2 protein)
TNF-α	↑↑	Activated macrophages, monocytes, B cells, T cells, lymphocytes, NK cells, polymorphonuclear leukocytes, eosinophils, astrocytes, Langerhans cells, Kupffer cells, glial cells, and adipocytes.	All nucleated cells	TNFR1, TNFR2	TNF-α inhibitor (Adalimumab)
IL-1β	↑↑	Blood monocytes, tissue macrophages, skin dendritic cells, and brain microglia.	Smooth muscle cells, Type 3 innate lymphoid cells, γδT cells, Th17 cells, monocytes, macrophages, epithelial, endothelial cells, chondrocytes, and fibroblasts.	IL-1R	IL-1β monoclonal antibody (Canakinumab), IL-1 receptor antagonist blocking IL-1α and IL-1β (Anakinra)
CXCL2	↑	Neutrophils	Neutrophils	CXCR2	CXCR2 blocker (Danirixin)
CCL2/5	↑	Macrophages, monocytes, and dendritic cells.	Macrophages, monocytes, Treg cells, and dendritic cells.	CCR2, CCR5	CCR2 blocker (Prozalizumab), CCL2 monoclonal antibody (Carlumab), CCR5 monoclonal antibody receptor antagonist (Leronlimab)

## Data Availability

Not applicable.

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
