# Peer review of "Emerging Effects of IL-33 on COVID-19"

_ijms, 2022, doi:10.3390/ijms232113656_

Round 1
Reviewer 1 Report
Gao et al describe in much detail the role of IL-33 in the severe pathology of COVID-19.
Whereas the role of this cytokine as one of the various members in a so-called cytokine storm as a result of the infection is well known. What is missing is the possible interplay with other molecules.
It would be very helpful to indicate the various pathways known today in dealing with COVID-19 infection. This could be done in a Figure similar to No. 2. In particular, the immediate induction of type I and III interferon (IFN) is crucial to control COVID-19 infections (1). Preexisting antibodies to type I IFN further support the role of this cytokine (2), (3). Plasmacytoid dendritic cells sensing the virus may play essential parts (4).
While rapid induction of type I IFNs limits virus propagation, sustained elevation of type I IFNs in the late phase of the infection is associated with aberrant inflammation and poor clinical outcome (5). This context would explain the facts, that younger healthy people may experience no or very mild symptoms, while elderlies with increased autoantibodies to type I IFN and limited cell numbers essential to immediately produce this cytokine might be at higher risk to experience lung pathology at least for the Wuhan strain.
1. Greene TT, Zuniga EI. 2021. Type I Interferon Induction and Exhaustion during Viral Infection: Plasmacytoid Dendritic Cells and Emerging COVID-19 Findings. Viruses 13.
2. Bastard P, Orlova E, Sozaeva L, Levy R, James A, Schmitt MM, Ochoa S, Kareva M, Rodina Y, Gervais A, Le Voyer T, Rosain J, Philippot Q, Neehus AL, Shaw E, Migaud M, Bizien L, Ekwall O, Berg S, Beccuti G, Ghizzoni L, Thiriez G, Pavot A, Goujard C, Fremond ML, Carter E, Rothenbuhler A, Linglart A, Mignot B, Comte A, Cheikh N, Hermine O, Breivik L, Husebye ES, Humbert S, Rohrlich P, Coaquette A, Vuoto F, Faure K, Mahlaoui N, Kotnik P, Battelino T, Trebusak Podkrajsek K, Kisand K, Ferre EMN, DiMaggio T, Rosen LB, Burbelo PD, McIntyre M, Kann NY, et al. 2021. Preexisting autoantibodies to type I IFNs underlie critical COVID-19 pneumonia in patients with APS-1. J Exp Med 218.
3. Manry J, Bastard P, Gervais A, Le Voyer T, Rosain J, Philippot Q, Michailidis E, Hoffmann HH, Eto S, Garcia-Prat M, Bizien L, Parra-Martinez A, Yang R, Haljasmagi L, Migaud M, Sarekannu K, Maslovskaja J, de Prost N, Tandjaoui-Lambiotte Y, Luyt CE, Amador-Borrero B, Gaudet A, Poissy J, Morel P, Richard P, Cognasse F, Troya J, Trouillet-Assant S, Belot A, Saker K, Garcon P, Riviere JG, Lagier JC, Gentile S, Rosen LB, Shaw E, Morio T, Tanaka J, Dalmau D, Tharaux PL, Sene D, Stepanian A, Megarbane B, Triantafyllia V, Fekkar A, Heath JR, Franco JL, Anaya JM, Sole-Violan J, Imberti L, et al. 2022. The risk of COVID-19 death is much greater and age dependent with type I IFN autoantibodies. Proc Natl Acad Sci U S A 119:e2200413119.
4. Becker J, Kalinke U. 2022. Toll-like receptors matter: plasmacytoid dendritic cells in COVID-19. EMBO J 41:e111208.
5. Di Domizio J, Gulen MF, Saidoune F, Thacker VV, Yatim A, Sharma K, Nass T, Guenova E, Schaller M, Conrad C, Goepfert C, De Leval L, von Garnier C, Berezowska S, Dubois A, Gilliet M, Ablasser A. 2022. The cGAS-STING pathway drives type I IFN immunopathology in COVID-19. Nature doi:10.1038/s41586-022-04421-w.
Author Response
Dear reviewers,
Thank you for your kind comments on our manuscript (ijms-1949848) entitled “Emerging Effects of IL-33 on COVID-19”. We appreciate the time and effort that you dedicated to providing feedback on our manuscript and are grateful for the insightful comments on and valuable improvements to our paper. According to your comments, we have carefully revised the manuscript again and made extensive modifications to the original manuscript. Our responses are given in a point-by-point manner below. Changes to the manuscript are marked in red.
Response to Reviewer 1 Comments
Point 1: Gao et al describe in much detail the role of IL-33 in the severe pathology of COVID-19.
Whereas the role of this cytokine as one of the various members in a so-called cytokine storm as a result of the infection is well known. What is missing is the possible interplay with other molecules.
It would be very helpful to indicate the various pathways known today in dealing with COVID-19 infection. This could be done in a Figure similar to No. 2. In particular, the immediate induction of type I and III interferon (IFN) is crucial to control COVID-19 infections (1). Preexisting antibodies to type I IFN further support the role of this cytokine (2), (3). Plasmacytoid dendritic cells sensing the virus may play essential parts (4).
While rapid induction of type I IFNs limits virus propagation, sustained elevation of type I IFNs in the late phase of the infection is associated with aberrant inflammation and poor clinical outcome (5). This context would explain the facts, that younger healthy people may experience no or very mild symptoms, while elderlies with increased autoantibodies to type I IFN and limited cell numbers essential to immediately produce this cytokine might be at higher risk to experience lung pathology at least for the Wuhan strain.
Responses: Thank you for your kind help and constructive suggestions. The core point of this paper is to elucidate the direct or indirect interaction between IL-33 and various molecules in the pathogenesis and development of COVID-19 by targeting various immune cells, aiming to provide new molecular targets for the prevention, diagnosis, and treatment of COVID-19. Although IL-33, as one of the key inflammatory molecules for the cytokine storm, plays a key role in the pathogenesis and development of COVID-19, the interaction and the underlying regulatory mechanisms between IL-33 and other molecules are still unclear. As you commented, interferon (IFN)-I, a cytokine derived from plasmacytoid dendritic cells (pDC), plays a key role in the pathogenesis of COVID-19. In the early stages of COVID-19 pathological development, the immediate induction of type I and III IFNs significantly limits the spread of the virus. However, pre-existing autoantibodies that neutralize type I IFN in APS-1 patients are associated with abnormal inflammation and poor clinical outcomes at any age. Clinically, young healthy individuals without pre-existing type I IFN antibodies may show no clinical symptoms or very mild symptoms when first infected with SARS-CoV-2. On the contrary, elderly individuals with increased autoantibodies to type I IFN and limited cell numbers essential to immediately produce this cytokine might be at higher risk to experience lung pathology, at least for the Wuhan strain. Previous studies have confirmed that IL-33 can effectively regulate dendritic cell proliferation and IFN-I expression and release(1). However, the underlying molecular mechanism by which IL-33 regulates IFN-I expression in COVID-19 remains unknown. Based on your valuable suggestion, Figure 4 is supplemented in the revised manuscript, aiming to provide a relatively comprehensive overview of the IL-33-interferon interaction signaling pathway using various immune cells as a bridge, especially pDC. Please refer to Figure 4 and the revised manuscript for details.
The detailed revision was shown as follows:
Dendritic cells (DCs), as efficient professional antigen-presenting cells responsible for bridging the gap between innate and adaptive immunity, are essential in recognizing pathogens and secreting inflammatory mediators(2, 3). DCs arise from bone marrow progenitors known as common myeloid progenitors (CMPs). Dendritic cells can be divided into two subpopulations based on the expression of surface markers: conventional dendritic cells (cDCs) and plasmacytoid dendritic cells (pDCs)(4-6). cDCs are primarily responsible for presenting antigens and stimulating naive T cell expansion and differentiation. pDCs typically produce high amounts of type I IFN (IFN-I) and pro-inflammatory cytokines including IL-6 and TNF-α in response to in vivo RNA and DNA viral infection (7-10). IL-6, IL-1β, TNF-α, and IFN-I were released by pDCs, which are key factors in cytokine storm and correlate with disease severity. According to current authoritative journal reports, the cytokine storm generated by the activation of various inflammatory signaling pathways is the hallmark pathological change of COVID-19 (11, 12). Following a pathogen attack, activation of immune cells (T cells, dendritic cells, macrophages, natural killer cells, and cytotoxic lymphocytes) occurs, resulting in the release of cytokines and chemokines and subsequently leading to an inflammatory response for viral clearance. Cytokines, in turn, may act on different cells, nearby cells, distant cells, as well as on the cell that secretes them. Notably, in the initial stage, the moderate release of cytokines exhibits beneficial inflammatory effects, acting only on viral cells but after the over-activation of the immune system, the over-produced cytokines also rapidly kill the host cells (13). However, the interaction mechanisms between immune cells, especially pDCs and cytokines, as well as between cytokines themselves in the pathogenesis and development of COVID-19 remain incompletely characterized.
Interferon, one of the most important antiviral cytokines, whose function declines or defects can lead to reduced resistance to viral infections, including but not limited to COVID-19 (14-16). Type I interferon (IFN-I) mainly includes three subtypes: IFN-α, β, and ω, which can participate in the antiviral response of immune cells through the JAK-STAT signaling pathway and the induction of interferon-stimulated genes (ISGs)(17). Numerous studies have shown that impaired IFN-I responses are associated with severe and critical COVID-19 infections. For instance, pre-existing autoantibodies to IFN-I in COVID-19 patients with APS-1 can induce severe pneumonia (18, 19). Therefore, rapid induction of IFN-1 production in vivo can effectively control early pathological damage after COVID-19 infection (20). Although IFN-I plays a protective role in the early stage of COVID-19 infection, the continuous increase of IFN-I levels can exacerbate abnormal inflammatory responses and aggravate pathological damage through the cGAS-STING pathway in the late stage of COVID-19 infection (21). As is well known, IFN-I can be produced in almost all cell types, but pDCs have been recognized as the main source of IFN-I because the production of IFN-I depends on the toll-like receptor-7 (TLR7)/TLR9 pathway, of which TLR7 and TLR9 are specifically expressed in pDCs(22). Upon COVID-19 infection, TLR7 or TLR9 activates MyD88 and IL-1 receptor-associated kinase 4 (IRAK-4), which then interact with tumor necrosis factor receptor-associated factor-6 (TRAF6), TRAF3, IRAK1, IKKα, and interferon regulatory factor 7 (IRF7) interaction. Ultimately, IRAK-1 and IKKα phosphorylate IRF7, leading to IRF7 activation and induction of IFN-I gene transcription and massive IFN-I production(23, 24). Although the reduction of IFN-I production by pDCs is likely to be an important factor in the pathological aggravation of COVID-19, the specific molecular mechanism of their mutual regulation remains unclear.
IL-33, as a major member of the cytokine storm, plays a key role in the pathological changes after COVID-19. However, whether the interaction mechanism of IL-33 with pDC or IFN-I is involved in the pathological development of COVID-19 remains unknown. Previous studies found that IL-33 treatment significantly limited IFN-α production, at least in part, by down-regulating the secretory function of pDCs(1,25). Specifically, IL-33 affects TLR7-mediated pDC activation by rapidly depleting the intracellular adaptor molecules IRAK1 and viperin, resulting in a hyporesponsive state of TLR7. The ability of IL-33 to reduce the secretion of IFN-I by pDCs is dependent on the specific receptor ST2 localized on the surface of pDCs(1,25). Since the TLR7 pathway is required for recognition of the SARS-CoV-2 genome and production of IFN-I, the IL-33/ST2 axis may suppress innate antiviral immunity and delay viral clearance in COVID-19 patients by reducing IFN-I expression (Figure 4). Additionally, an IFN-α/λlow IL-33high cytokine microenvironment allows for the onset of type 2 immune response and early viral growth(1). This in turn results in persistent changes in alveolar epithelial cells and immune cells, which are the immune basis of severe lung injury following COVID-19 infection. Collectively, the IL-33/ST2 axis remarkably reduces pDCs-dependent IFN-I production by inhibiting the biological activity of the IRAK1/TRL7 pathway, thereby increasing the susceptibility of COVID-19 patients to the virus in the early to middle stage, especially for patients with insufficient circulating IFN levels. This may well explain the clinically milder and more self-limiting symptoms in younger patients and more severe systemic pathological reactions and complications in older patients. Therefore, targeting the intervention of IL-33/ST2/pDCs/IFN-I signaling pathway may provide new therapeutic opportunities for COVID-19.
References
- Lynch J P, Werder R B, Simpson J, et al. Aeroallergen-induced IL-33 predisposes to respiratory virus-induced asthma by dampening antiviral immunity[J]. J Allergy Clin Immunol, 2016, 138(5): 1326-1337.
- Schuijs M J, Hammad H, Lambrecht B N. Professional and 'Amateur' Antigen-Presenting Cells In Type 2 Immunity[J]. Trends Immunol, 2019, 40(1): 22-34.
- Morante-Palacios O, Fondelli F, Ballestar E, et al. Tolerogenic Dendritic Cells in Autoimmunity and Inflammatory Diseases[J]. Trends Immunol, 2021, 42(1): 59-75.
- Villar J, Segura E. Decoding the Heterogeneity of Human Dendritic Cell Subsets[J]. Trends Immunol, 2020, 41(12): 1062-1071.
- Macri C, Pang E S, Patton T, et al. Dendritic cell subsets[J]. Semin Cell Dev Biol, 2018, 84: 11-21.
- Worbs T, Hammerschmidt S I, Förster R. Dendritic cell migration in health and disease[J]. Nat Rev Immunol, 2017, 17(1): 30-48.
- Li Q, Liu Y, Wang X, et al. Regulation of Th1/Th2 and Th17/Treg by pDC/mDC imbalance in primary immune thrombocytopenia[J]. Exp Biol Med (Maywood), 2021, 246(15): 1688-1697.
- Reizis B. Plasmacytoid Dendritic Cells: Development, Regulation, and Function[J]. Immunity, 2019, 50(1): 37-50.
- Swiecki M, Colonna M. The multifaceted biology of plasmacytoid dendritic cells[J]. Nat Rev Immunol, 2015, 15(8): 471-85.
- Reizis B, Bunin A, Ghosh H S, et al. Plasmacytoid dendritic cells: recent progress and open questions[J]. Annu Rev Immunol, 2011, 29: 163-83.
- Ramasamy S, Subbian S. Critical Determinants of Cytokine Storm and Type I Interferon Response in COVID-19 Pathogenesis[J]. Clin Microbiol Rev, 2021, 34(3).
- England J T, Abdulla A, Biggs C M, et al. Weathering the COVID-19 storm: Lessons from hematologic cytokine syndromes[J]. Blood Rev, 2021, 45: 100707.
- Rodríguez Y, Novelli L, Rojas M, et al. Autoinflammatory and autoimmune conditions at the crossroad of COVID-19[J]. J Autoimmun, 2020, 114: 102506.
- Samuel C E. Antiviral actions of interferons[J]. Clin Microbiol Rev, 2001, 14(4): 778-809, table of contents.
- Mcnab F, Mayer-Barber K, Sher A, et al. Type I interferons in infectious disease[J]. Nat Rev Immunol, 2015, 15(2): 87-103.
- Lei X, Dong X, Ma R, et al. Activation and evasion of type I interferon responses by SARS-CoV-2[J]. Nat Commun, 2020, 11(1): 3810.
- Liu S Y, Sanchez D J, Aliyari R, et al. Systematic identification of type I and type II interferon-induced antiviral factors[J]. Proc Natl Acad Sci U S A, 2012, 109(11): 4239-44.
- Manry J, Bastard P, Gervais A, et al. The risk of COVID-19 death is much greater and age dependent with type I IFN autoantibodies[J]. Proc Natl Acad Sci U S A, 2022, 119(21): e2200413119.
- Bastard P, Orlova E, Sozaeva L, et al. Preexisting autoantibodies to type I IFNs underlie critical COVID-19 pneumonia in patients with APS-1[J]. J Exp Med, 2021, 218(7).
- Sodeifian F, Nikfarjam M, Kian N, et al. The role of type I interferon in the treatment of COVID-19[J]. J Med Virol, 2022, 94(1): 63-81.
- Domizio J D, Gulen M F, Saidoune F, et al. The cGAS-STING pathway drives type I IFN immunopathology in COVID-19[J]. Nature, 2022, 603(7899): 145-151.
- Greene T T, Zuniga E I. Type I Interferon Induction and Exhaustion during Viral Infection: Plasmacytoid Dendritic Cells and Emerging COVID-19 Findings[J]. Viruses, 2021, 13(9).
- Blasius A L, Beutler B. Intracellular toll-like receptors[J]. Immunity, 2010, 32(3): 305-15.
- Kawai T, Akira S. Toll-like receptors and their crosstalk with other innate receptors in infection and immunity[J]. Immunity, 2011, 34(5): 637-50.
- Wu M, Gao L, He M, et al. Plasmacytoid dendritic cell deficiency in neonates enhances allergic airway inflammation via reduced production of IFN-α[J]. Cell Mol Immunol, 2020, 17(5): 519-532.

Reviewer 2 Report
Authors provided very comprehensive, illustrative and informative review about the immunology of cytokine storm in COVID-19, with the focus on IL-33. Literature which is used is relevant for the subject, and completely accurate. I did not identify any gaps in knowledge. By describing in detail pathophysiology of the IL-33 itself at the beginning, and further explaining its potential practical role in everyday practice as predictive factor for severe COVID-19 infection, which opens ideas for further clinical trials, as presented in Table 2. Table 2 is really informative, as well as all other figures. According to my opinion, authors responded correctly to the aim of the review, so my recommendation would be acceptance in present form.
Author Response
Dear reviewers,
Thank you for your kind comments on our manuscript (ijms-1949848) entitled “Emerging Effects of IL-33 on COVID-19”. We appreciate the time and effort that you dedicated to providing feedback on our manuscript and are grateful for the insightful comments on and valuable improvements to our paper. According to your comments, we have carefully revised the manuscript again and made extensive modifications to the original manuscript. Our responses are given in a point-by-point manner below. Changes to the manuscript are marked in red. The detailed revision was shown as follows.
Response to Reviewer 2 Comments
Point 1: Authors provided very comprehensive, illustrative and informative review about the immunology of cytokine storm in COVID-19, with the focus on IL-33. Literature which is used is relevant for the subject, and completely accurate. I did not identify any gaps in knowledge. By describing in detail pathophysiology of the IL-33 itself at the beginning, and further explaining its potential practical role in everyday practice as predictive factor for severe COVID-19 infection, which opens ideas for further clinical trials, as presented in Table 2. Table 2 is really informative, as well as all other figures. According to my opinion, authors responded correctly to the aim of the review, so my recommendation would be acceptance in present form.
Response : Thanks for the reviewer’s rigorous and serious work attitude.

Reviewer 3 Report
In this presented manuscript, Gao et al. focused on discussing the immune response and cytokine storm stage of COVID-19 pathology. The authors first summarized several cytokines and chemokines that are upregulated in the cytokine storm of COVID-19, and then chose IL-33 as the primary target to evaluate. The initial background introduction of IL-33 and the IL-33/ST2 signaling pathway helped the audience to follow the later analysis. The main discussions of the manuscript were about how IL-33 regulates the immune cells involved in the development of COVID-19 pathology. The authors comprehensively summarized how IL-33 enhances neutrophils, stimulates innate lymphoid type-2 cells, activates macrophages, regulates CD4+ T cells, mediates Th17/Treg, and promotes CD8+ T cells. All the above interpretation supports the speculation that IL-33 could play an important role in pathological inflammation formation through immune cell regulation and cytokine regulation in COVID-19. This review provides insights into the clinal prognosis and treatment of COVID-19 patients.
Some minor suggestions regarding the manuscript figures:
Figure 1 (c): the text color of ST2 domains is not very visible under the pale yellow background
Figure 2 and Figure 3: may want to use pictures with a higher resolution
Author Response
Dear reviewers,
Thank you for your kind comments on our manuscript (ijms-1949848) entitled “Emerging Effects of IL-33 on COVID-19”. We appreciate the time and effort that you dedicated to providing feedback on our manuscript and are grateful for the insightful comments on and valuable improvements to our paper. According to your comments, we have carefully revised the manuscript again and made extensive modifications to the original manuscript. Our responses are given in a point-by-point manner below. Changes to the manuscript are marked in red. The detailed revision was shown as follows.
Response to Reviewer 3 Comments
In this presented manuscript, Gao et al. focused on discussing the immune response and cytokine storm stage of COVID-19 pathology. The authors first summarized several cytokines and chemokines that are upregulated in the cytokine storm of COVID-19, and then chose IL-33 as the primary target to evaluate. The initial background introduction of IL-33 and the IL-33/ST2 signaling pathway helped the audience to follow the later analysis. The main discussions of the manuscript were about how IL-33 regulates the immune cells involved in the development of COVID-19 pathology. The authors comprehensively summarized how IL-33 enhances neutrophils, stimulates innate lymphoid type-2 cells, activates macrophages, regulates CD4+ T cells, mediates Th17/Treg, and promotes CD8+ T cells. All the above interpretation supports the speculation that IL-33 could play an important role in pathological inflammation formation through immune cell regulation and cytokine regulation in COVID-19. This review provides insights into the clinal prognosis and treatment of COVID-19 patients.
Point 1: Figure 1 (c): the text color of ST2 domains is not very visible under the pale yellow background.
Response: We appreciate the reviewers' rigorous and serious work attitude, and hope that the correction will meet with approval. The background color of Figure 1 is replaced by blue, and the color of the "Domain" text in Figure 1(c) is replaced by black, which makes Figure 1 distinct and the content more clearly visible. Please refer to Figure 1 in the revised manuscript for details.
Point 2: Figure 2 and Figure 3: may want to use pictures with a higher resolution.
Response: Thank you for your helpful and valuable suggestions. Per the reviewer’s suggestions, Figures 2 and 3 have been replaced with high-resolution figures, please refer to Figures 2 and 3 in the revised manuscript for details.

Round 2
Reviewer 1 Report
Excellent response that is very valuable and puts IL-33 in the immune context. The figures are very clearly done. Congratulations